# Recent Advances in Light Penetration Depth for Postharvest Quality Evaluation of Fruits and Vegetables

**DOI:** 10.3390/foods13172688

**Published:** 2024-08-26

**Authors:** Yuping Huang, Jie Xiong, Ziang Li, Dong Hu, Ye Sun, Haojun Jin, Huichun Zhang, Huimin Fang

**Affiliations:** 1College of Mechanical and Electronic Engineering, Nanjing Forestry University, Nanjing 210037, China; xiongjie@njfu.edu.cn (J.X.); 13805524100@163.com (Z.L.); njzhanghc@hotmail.com (H.Z.); 2College of Optical, Mechanical and Electrical Engineering, Zhejiang A&F University, Hangzhou 311300, China; hudong538338@zju.edu.cn; 3College of Food Science and Light Industry, Nanjing Tech University, Nanjing 211816, China; sunye@njau.edu.cn; 4School of Flexible Electronics (Future Technologies) and Institute of Advanced Materials (IAM), Nanjing Tech University, Nanjing 211816, China; iamhjjin@njtech.edu.cn; 5School of Agricultural Engineering, Jiangsu University, Zhenjiang 212013, China; fanghuimin@ujs.edu.cn

**Keywords:** light penetration depth, optical properties, quality evaluation, optical detection techniques, applications

## Abstract

Light penetration depth, as a characteristic parameter reflecting light attenuation and transmission in biological tissues, has been applied in nondestructive detection of fruits and vegetables. Recently, with emergence of new optical detection technologies, researchers have begun to explore methods evaluating optical properties of double-layer or even multilayer fruit and vegetable tissues due to the differences between peel and pulp in the chemical composition and physical properties, which has gradually promoted studies on light penetration depth. A series of demonstrated research on light penetration depth could ensure the accuracy of the optical information obtained from each layer of tissue, which is beneficial to enhance detection accuracy for quality assessment of fruits and vegetables. Therefore, the aim of this review is to give detailed outlines about the theory and principle of light penetration depth based on several emerging optical detection technologies and to focus primarily on its applications in the field of quality evaluation of fruits and vegetables, its future applicability in fruits and vegetables and the challenges it may face in the future.

## 1. Introduction

As the problem of food safety has attracted increasing attention, it is of great practical importance to develop efficient and nondestructive techniques to ensure consumers’ acceptance and improve the accuracy of postharvest grading of fruits and vegetables [1,2]. Up to now, a series of emerging techniques based on optical properties, acoustic characteristics, dielectric properties and magnetism have been developed, which can obtain information about the quality without damaging the chemical and physical properties of the foods. The acoustic technique is often used for the detection of firmness, sugar percentage and maturity [3] with low cost and simple detection equipment but low detection accuracy. The dielectric technique, e.g., microwave imaging [4], is capable of judging the firmness, maturity, sugar percentage and other quality parameters [5]. However, it has not reached the practical stage due to the factors (i.e., determination of frequency/voltage signal, ambient temperature and humidity) affecting the detection accuracy. On the basis of the principle of nuclear magnetism, the nuclear magnetic resonance (NMR) technique has strong penetration and is qualified for evaluating the internal defects of vegetables and fruits [6], but it is unsuitable for the practical application of agricultural products due to the high cost of the equipment. Thanks to the limitations of the above techniques, researchers have begun to assess quality attributes through optical properties of biological tissues and corresponding detection techniques. Compared with other detection techniques, optical detection techniques, such as hyperspectral imaging spectroscopy [7,8] and visible/near-infrared spectroscopy techniques [9,10], are widely used to evaluate quality attributes of fruits and vegetables, e.g., soluble solid content [11], maturity [12], defects [13], acidity [14], varieties [15] and safety [16]. In addition, some novel optical techniques emerged with the development of optoelectronic techniques, such as time-resolved spectroscopy (TRS) [17,18], spatial frequency domain imaging (SFDI) [19,20] and structured light reflection imaging (SIRI) [21].

Optical detection techniques are mainly based on the principle of interaction between light and biological tissues and use light reflection, transmission and semi-transmission to obtain physical and chemical information without destroying the sample, as shown in Figure 1. When illumination passes through the biological tissues, which here refer to the tissues of fruits and vegetables, scattering and absorption effects occur during the process [22]. Absorption is mainly described by the absorption coefficient ($\mu_{a}$), which depends on the chemical compositions, such as chlorophyll, while scattering is largely represented by the scattering coefficient ($\mu_{s}$), which is in connection with physical properties, e.g., cell size and cellular structure [23]. Other optical parameters could be derived from two basic parameters ($\mu_{a}$ and $\mu_{s}$) according to a simplified model of radiative transfer theory, such as the penetration depth (δ), effective attenuation coefficient ($\mu_{\mathrm{eff}}$) and mean free path (mfp’). $\mu_{\mathrm{eff}}$ is a comprehensive parameter (related to $\mu_{a}$, $\mu_{s}$ and δ) describing the light absorption in the medium, which is generated by the interaction between structure inhomogeneity at the microscopic level and tissue chromophore concentration, while mfp’ represents the average distance for a photon travelling between two absorbing particles before being absorbed [21]. These parameters are helpful to further study optical properties in fruit and vegetable tissues. Most fruits and vegetables should be regarded as having a double or multilayer structure due to the existence of pericarp, leading to an increment of parameters involved in the optical transmission model and difficulty to accurately obtain the optical characteristic parameters [24]. In order to simplify the problem, many researchers normally consider fruits and vegetables as single homogeneous samples when measuring their optical properties. However, for some fruits and vegetables with thicker pericarp, such as citrus and mangoes, there is a big difference between pericarp and pulp tissues involving chemical composition and physical properties, and the results of such a method will produce large errors, which means that it is difficult to evaluate the quality of these products with thicker pericarp. To solve this problem, on the basis of spatially resolved diffuse reflectance spectroscopy, Wang et al. [25] adopted a sequential method to measure the optical properties of double-layer tissue, finding that light penetration depth enhanced with increasing source-detector separation (SDS) in double-layered media [26].

Since the 1980s, the study of light penetration depth in the tissues of fruits and vegetables has been reported. Researchers have defined light penetration depth in a variety of ways for different studies. According to the diffusion theory, Wilson and Jacques [27] defined the light penetration depth as the propagation distance when illumination intensity is decreased to 1/e (about 37%). Lammertyn et al. [28] defined light penetration depth as slice thickness, where a diffuse spectrum of light differs significantly from that of slices with infinite thickness. In their research, light penetration depth was determined by the spectra obtained from apple slices of different thicknesses. Moreover, the diffuse reflectance spectra at different thicknesses varies with that collected at infinite thicknesses. In another study, light penetration depth was proposed by Fraser as the depth reached when illumination intensity is up to 1% [29]. The penetration depth profile obtained was similar to that acquired by Lammertyn et al. [28], but the results were quite different. Later, in terms of blood glucose concentration measurement, the penetration depth was the maximum longitudinal position reached by diffuse photons in tissues [26]. Combined with the photon density function and Monte Carlo Multi-Layered (MCML) simulation, penetration depth distribution of photons was discussed, and the relationship between the average penetration depth and SDS was determined. Although these definitions of light penetration depth are different, they are all based on the evolution of the radiation transport equation. Wilson and Jacques [27] and Fraser [29] defined the light penetration depth based on the diffuse approximate equation, while Sujatha et al. [26] calculated the photon penetration depth by describing the motion track of the light transmission process through MCML simulation. The diffuse approximate equation is one of the most commonly used simplified models of a radiation transfer equation, while the Monte Carlo method is a random statistical method. In addition, the adding-doubling method is also a common numerical method for resolving the radiative transfer equation. The study of light penetration depth is not only favored for obtaining longitudinal information about various hierarchies in tissues, but it can also directly reflect the attenuation of light in tissues [30], which is significant for determining the best sensing position of the sample tissue. It is noteworthy that most studies only reported on light penetration depth in specific sample tissues [31,32,33], while fewer reviews of light penetration depth in fruits and vegetables were conducted. The summary of explanations of light penetration depth and its application to fruits and vegetables will help scholars better understand the theory of light transmission.

Therefore, this paper reviews recent advances on light penetration depth in the quality evaluation of fruits and vegetables, involving the definition and means of light penetration depth as well as related influencing factors. The theories of optical properties (absorption, scattering and light penetration depth) and research methods of light transmission are introduced, and then the working principles and developments of several optical detection techniques are briefly overviewed. Following this, applications of these techniques to the quality evaluation of fruits and vegetables, which focuses on light penetration depth, are reviewed. Finally, the challenges of studying light penetration depth are discussed along with perspectives.

## 2. The Theory of Absorption, Scattering and Penetration Depth

Light transmission in turbidity or diffusion in biological tissue is a complex phenomenon, involving absorption and scattering represented by three basic parameters: absorption coefficient, scattering coefficient and anisotropy factor. Common parameters such as the effective attenuation coefficient and light penetration depth could also be expressed by three basis parameters, as shown in Equations (6) and (7). The absorption and scattering are the principal factors that determine the interaction between light and turbid media, in connection with the chemical composition and physical properties of biological tissue, respectively [34].

Absorption refers to when photons are annihilated by atoms, electrons, etc. and converted into other forms, such as chemical, electrical or heat energy. The absorption properties could be portrayed by the absorption coefficient, and thus the absorption coefficient is introduced as a quantization of luminous energy transformed into other forms, which is defined as the probability of absorption of photons after propagation of unit length. The absorption coefficient ($\mu_{a}$) can be calculated by the Beer–Lambert law [35]:(1)$$I=I_{0}\exp\left(-\mu_{a}\cdot d\right)$$
where I is the transmitted light, $I_{0}$ is the incident intensity and *d* is the thickness of the pure absorbing medium. Since the absorption of biological tissues depends on chemical composition, the absorption coefficient is generally used to establish the relationship between absorption and chemical composition, so as to achieve the purpose of evaluating the internal quality of objects.

Scattering results from changes in the direction of photon propagation when they hit the particles, due to the discrete particles and the difference in the refractive index. In terms of scattering in biological tissues, it could be divided into inelastic and elastic scattering. Rayleigh and Mie scattering belong to elastic scattering, while Raman scattering is inelastic scattering. To be specific, Rayleigh scattering arises from electrical polarization of particles and occurs when photons come into contact with smaller particles [36]. Therefore, Rayleigh scattering is only appropriate for the ultrastructure. Mie scattering is applicable to particles whose size is similar to optical waves, while cells in tissues are mostly about a few microns in size [37]. Hence, the Mie scattering theory can be utilized to portray the light scattering phenomenon in tissues. When Mie scattering occurs, the photon path becomes less direct. Based on this, scholars introduced the concept of the scattering coefficient ($\mu_{s}$), which quantifies photon scattering probability [38] and is applied to the Beer–Lambert law:(2)$$I=I_{0}\exp\left(-\mu_{s}\cdot d\right)$$
where I is the transmitted light, and *d* is the thickness of the scattering medium. In biological tissues, light scattering is uneven in all directions. By introducing the anisotropy coefficient (g) and the combining scattering coefficient to obtain the reduced scattering coefficient ($\mu {’}_{s}$), the variables of the transmission model could be reduced [39]:(3)$$\mu {’}_{s}=\mu_{s}(1-g)$$
where the value of g ranges from −1 to 1, and mfp’ could be represented by $\mu_{a}$ and $\mu {’}_{s}$ as $mfp’=1/(\mu_{a}+\mu {’}_{s})$. Raman scattering occurs when a strong beam of light hits a molecule. When the excited molecule relaxes immediately, inelastic scattered photons are produced. According to the relationship between energy and frequency, two frequencies of scattered light are finally formed, namely Stokes scattered light and Anti-Stokes scattered light [40]. If the molecule gets energy from the incident photon during the collision to make it transition from a lower energy level to a higher energy level, the frequency of the photon will drop after the collision, which results in Stokes scattering, while Anti-Stokes scattering occurs when part of the energy is transferred to the incident photon in the collision, making it transition from a higher energy level to a lower energy level, and the photon frequency increases, as shown in Figure 2, which also illustrates the energy change of Rayleigh scattering and Raman scattering during the scattering process. The light emitted by Raman scattering differs from the excited light in energy and wavelength. The Raman spectrum was generated based on this principle to assess the quality of fruits and vegetables. To sum up, the Mie scattering theory can be used to explain the light scattering in tissues for fruit and vegetables.

The $\mu_{\mathrm{eff}}$ and penetration depth (δ) are related to the absorption and scattering [41]. Information about the inner parts of biological tissues could be obtained effectively by the determination of light penetration depth. According to the diffusion theory, luminous intensity in turbid media reduces exponentially with $\mu_{\mathrm{eff}}$ as the attenuation constant:(4)$$I=I_{0}\exp(-\mu_{eff}\cdot L)$$
where *I* represents the transmitted light entering a specific depth of tissues, *I*_0_ represents luminous intensity entering the surface and L is the depth entering tissues from a measured place nearest the edge of the light. According to $\mu_{eff}=1/\delta$, the expression of the relation between light intensity and penetration depth can be obtained as follows:(5)$$I=I_{0}\exp(-L/\delta )$$
according to Wilson’s definition of penetration depth, when *I*/*I*_0_ = 1/*e* ≈ 37%, *δ* = *L*. According to Fraser’s description, the light penetration depth is defined as the distance reached when the light intensity drops to 1%, i.e., *I*/*I*_0_ = 1%, here *L* = 4.6 *δ*, as shown in Figure 3.

Under the condition of steady-state diffusion approximation, the δ depends on $\mu_{a}$ and $\mu {’}_{s}$ [32]:(6)$$\delta =\frac{1}{\mu_{eff}}=\frac{1}{\sqrt{3\mu_{a}\left(\mu_{a}+\mu {’}_{s}\right)}}$$
where $\mu {’}_{s}\gg \mu_{a}$. In the case of sinusoidal structured illumination, Equation (6) can be transformed as follows [42]:(7)$$\delta =\frac{1}{\mu {’}_{eff}}=\frac{1}{\sqrt{\mu_{eff}^{2}+{\left(2\pi f_{x}\right)}^{2}+{\left(2\pi f_{y}\right)}^{2}}}$$
where $f_{x}$ and $f_{y}$ represent spatial frequencies that are situated in the vertical and horizontal directions, respectively. Equations (6) and (7) show that under the conditions of the corresponding model, the δ could be directly calculated. For further exploration, some researchers have applied the Monte Carlo method (MC) to quantitatively analyze light transfer in biological tissues.

## 3. Light Transfer in Turbid Medium

The radiation transfer equation (RTE) could precisely depict the propagation of light in turbid media, but it usually needs to be solved under specific conditions due to the complexity of the model. At present, the common simplified models include the diffusion approximation equation, the Monte Carlo method, the adding-doubling method and so on. However, the diffusion approximation equation is widely applied for evaluation of optical properties of fruit and vegetables with the conditions of strongly scattering media and the SDS much larger than 1 mfp’ [38]. Due to the presence of biological chromophores, e.g., chlorophyll, carotenoids, etc., which have high absorption, the assumptions of the diffusion approximation equation are always inapplicable for some biological tissues [23]. To overcome these problems, the Monte Carlo method, the adding-doubling method, etc., are usually employed in simulations of light transmission in biological tissues. Table 1 summarizes the advantages and limitations of the above three simplified models, which helps to understand them directly and to ensure that the optimal simplified model could be selected to describe the transmission of light under specific conditions.

### 3.1. Diffusion Approximation Theory

Based on the law of conservation of energy, the RTE depicts the propagation of light in a medium [43]. However, it cannot directly calculate the analytical solution due to the complex model and the many independent variables involved in the equation. To solve this problem, when the medium is strongly scattering and the SDS is much larger than 1 mfp’ [23], the diffusion approximate equation could be expressed as a simplified model of the radiation approximate equation as follows:(8)$$\frac{\partial \phi (\vec{r},t)}{\partial t}=\nabla \cdot \left[D\nabla \phi (\vec{r},t)\right]-\mu_{a}\phi (\vec{r},t)+S(\vec{r},t)$$
where ∇· and ∇ are divergence and gradient operators in the three-dimensional space, respectively, $\phi (\vec{r},t)$ is the fluence rate, $\vec{r}$ is the location vector, $S(\vec{r},t)$ is the isotropic source in the medium and *D* is called the diffusion coefficient, which can be expressed as follows:(9)$$D=\frac{1}{3(\mu_{a}+\mu {’}_{s})}$$

At present, there are two commonly used analytical methods for this equation, which are both derived from the extrapolated boundary conditions. The solution of diffuse reflection from the surface of a semi-infinite medium was analyzed when an infinitesimal beam of light incident was upon it, as shown in Figure 4a. The equation is presented as follows [44,45]:(10)$$R(r)=\frac{a’}{4\pi }\left[z_{0}\left(\mu_{eff}+\frac{1}{r_{1}}\right)\frac{\exp\left(-\mu_{eff}r_{1}\right)}{r_{1}^{2}}+\left(z_{0}+2z_{b}\right)\left(\mu_{eff}+\frac{1}{r_{2}}\right)\frac{\exp\left(-\mu_{eff}r_{2}\right)}{r_{2}^{2}}\right]$$
where *r* is the SDS, *r*_1_ is the distance from the detector to the actual light source, *r*_2_ is the distance from the detector to the mirror light source, $A=(1+R_{f})/(1-R_{f})$ is the reflection coefficient inside tissues and *R_f_* is determined by the refractive index, $z_{0}={\left(\mu {’}_{t}\right)}^{-1}={\left(\mu_{a}+\mu {’}_{s}\right)}^{-1}$, $z_{b}=2AD$.

Based on the above model, researchers took the fluence rate into consideration and conveyed the diffuse reflectance as a linearity of fluence rate and flux [22,23]. In the diffusion approximate equation, the fluence rate can be expressed as
(11)$$\phi \left(r\right)=\frac{1}{4\pi D}\left(\frac{\exp\left(-\mu_{eff}r_{1}\right)}{r_{1}}-\frac{\exp\left(-\mu_{eff}r_{2}\right)}{r_{2}}\right)$$

The diffuse reflectance is converted as follows:(12)$$\begin{array}{c} R\left(r\right)=C_{1}\phi \left(r,z=0\right)+C_{2}\vec{J}\left(r,z=0\right)\\ =\frac{C_{1}}{4\pi D}\left[\frac{\exp\left(-\mu_{eff}r_{1}\right)}{r_{1}}-\frac{\exp\left(-\mu_{eff}r_{2}\right)}{r_{2}}\right]+\\ \frac{C_{2}}{4\pi }\left[z_{0}\left(\mu_{eff}+\frac{1}{r_{1}}\right)\frac{\exp\left(-\mu_{eff}r_{1}\right)}{r_{1}^{2}}+\left(z_{0}+2z_{b}\right)\left(\mu_{eff}+\frac{1}{r_{2}}\right)\frac{\exp\left(-\mu_{eff}r_{2}\right)}{r_{2}^{2}}\right] \end{array}$$
where $C_{1}=\frac{1}{4\pi }\int_{2\pi }\left[1-R_{fres}\left(\theta \right)\cos\theta d\omega \right]$ and $C_{2}=\frac{3}{4\pi }\int_{2\pi }\left[1-R_{fres}\left(\theta \right)\cos^{2}\theta d\omega \right]$ are the coefficients produced by the refractive index. The refractive index *n* is usually 1.35 for fruits and vegetables, while *C_1_* and *C_2_* are 0.1277 and 0.3269 [22], respectively. The analytical solution of Equation (10) has some errors because it directly regards the flux passing through the medium surface as the diffuse reflectance, while Equation (12) introduces the fluence rate, which could more precisely describe light transmission in biological materials.

### 3.2. Monte Carlo Method

The Monte Carlo (MC) method is a random statistical method, and photon propagation in turbid media could be described flexibly and precisely by MC simulations [25]. The MC model simulates the process of light transmission in biological tissues, including photon generation, transmission and reflection, absorption and scattering, random migration, etc. [46]. Meanwhile, it calculates the absorption and transmission of energy in tissues so as to obtain information related to light transmission in biological tissues, whose flow chart is shown in Figure 5a. In the 1970s, with the MC method introduced into the study of optics, the five basic steps of MC simulation were expounded. Subsequently, the MC method has been widely used for researches of propagation properties in biological tissues of two layers and multilayers, among which the classic model is the hybrid model [23], combining MC simulation with the diffusion approximation theory. Based on the C language, they developed a program that could be used to simulate multilayer tissues and finite light sources. Figure 4b shows diffuse light reflection and photon absorption in biological tissues under Lihong Wang’s simulation. In addition, a time-resolved MC model was developed to discuss the relationship between tissue optical parameters and time-resolved diffuse reflection [47]. The MC method has also been improved for retrieving optical parameters by using time-domain characteristics of optical propagation for biological tissues to enhance the computational efficiency [38]. Over the past few decades, MC methods have been widely used to model light propagation in different fields, such as biomedical science, food and agricultural fields.

In principle, MC simulation can simulate the transport of photons through biological tissues as long as the geometric structure and optical parameters of the tissue are determined. It simultaneously has good flexibility and robustness. However, it is unsuitable for rapid detection because of the large amount of calculation and long time for the simulation process.

### 3.3. Adding-Doubling Method

The adding-doubling (AD) method, a numerical method for solving the RTE, could deal with the boundary problem of refractive index change and the anisotropic scattering problem. The AD method, originated from the doubling method, was proposed to analyze the transmission process of light in the plate type sample [48]. As shown in Figure 4c, the emissivity of the reflected and transmitted light emitted at angle *v*′ from a light source incident at angle *v* could be obtained by numerical integration. The AD method is based on the following assumptions: a. no time dependence, b. the sample structure is an infinite plane of finite thickness, c. the optical properties of the layered structure are evenly distributed and d. the collimated or diffused light uniformly illuminates the surface.

To measure the optical parameters of biological tissues, an inverse adding-doubling (IAD) method based on the AD method was proposed, which is applied to calculate optical parameters, as shown in the flowchart in Figure 5b. The implementation of the IAD method relies on the reflectance (*R*) and transmittance (*T*), the anisotropy, the thickness, the refractive index of samples and so on [49]. Hu et al. [50] applied comparison measurements of the IAD method by changing the values of these parameters to determine the influence on the estimation of $\mu {’}_{s}$. Up to now, the IAD method is usually combined with the integrating sphere technique to calculate the optical properties of tissues. Lopez-Maestresalas et al. [51] applied the IAD method to extract optical properties for potatoes by using reflectance and transmittance spectra of potatoes over the spectral range of 500~1900 nm measured from a double-integrating sphere. Ma et al. [52] measured the optical properties of peaches within 600~1050 nm by a single integrating sphere system coupled with the IAD method to analyze the relationship between optical properties and firmness. Moreover, the IAD method was also employed in assessment of the optical properties for other fruits and vegetables such as pears [53], kiwifruits [54] and melons [55]. Compared with the MC method, the AD method has the same measurement accuracy and short calculation time. However, the AD method may take a long time for sample preparation due to the need for sample slices, and slicing may damage the tissue structure, causing changes in optical properties.

## 4. Optical Detection Techniques

The diffusion approximation equation has been solved analytically for several important illumination situations, including continuous-wave or steady-state illumination, pulsed point illumination, frequency-modulated illumination and spatially modulated area illumination. These analyses are deconstructed into the theoretical basis of time-resolved spectroscopy (TRS), spatially resolved spectroscopy (SRS) and frequency-domain (FD) and spatial-frequency domain imaging (SFDI). Table 2 summarizes the advantages and limitations of the mentioned optical detection techniques. In this section, TRS, SRS, FD, SFDI and even structured illumination reflectance imaging (SIRI) are introduced successively, all of which are applied for quality evaluation of fruits and vegetables.

### 4.1. Time Resolved Spectroscopy

As shown in Figure 6a, the principle of TRS for measuring optical properties could be summarized as follows: short laser pulses are injected into the mixed medium to record the time response function of the re-emitted photons at a fixed SDS, and then the optical properties of the medium are estimated by an inverse algorithm of the diffusion equation [56]. Based on the time delay and the broadening experienced by short laser pulses as they are passing through a turbid medium, TRS could estimate the $\mu_{a}$ and $\mu {’}_{s}$ by the photon distribution within the diffused medium [57]. Figure 6b illustrates the common configuration scheme of time-resolved spectra for measuring optical characteristics. Short laser pulses are transmitted to fruits or vegetables through a laser diode, and then re-emitted photons are collected by a photomultiplier at a fixed distance from the incident point. Subsequently, time-correlated single photon counting (TCSPC) is carried out on signals and ultimately summarized on the computer, in which the key components are the laser that generates the light source pulse and the TCSPC device.

In fact, since the 1990s, TCSPC has been applied to the TRS technique in biomedical applications. Over the years, the TRS system has gone through three generations of evolution, from the TCSPCY instrument based on the NIM module, to the TCSPCY instrument with a printed circuit board and then to the current TCSPCY device based on time-to-digital converters. It is easy to see that the TRS system developed gradually to have a high conversion rate, linearity and low cost [18]. In order to precisely achieve the optical properties of tissues, TCSPC equipment needs to provide temporal resolution in picoseconds (10^−12^ s) or even femtoseconds (10^−15^ s). In the early 2000s, a fully self-operating TRS system was first applied for the measurement of optical properties for several fruits in the red and near-infrared regions [58]. Subsequently, a pocket and low-cost portable prototype based on the TRS technique with the capability of automatic sorting or grading for apples was developed, which means that the TRS technique has the potential to analyze penetration depth. The TRS technique is normally used for the detection of internal quality for fruits and vegetables due to the advantage of insensitivity of surface features. Sarkar et al. [59] proposed a non-invasive method based on time-resolved phase reflectance to monitor the internal quality of apples. In addition, researchers in Italy developed three configurations of the TRS system to measure optical properties for horticultural products under different conditions (in a broad spectral range, at a single wavelength and at select discrete wavelengths) [17]. The instrument for TRS is expensive, even for a portable prototype. In order to realize smaller costs for detection, Sarkar et al. [60] established a time-resolved reflectance phase spectroscopy for nondestructive detection of fruits by using temporal resolution and phase information. The optical instruments used in this technique are simple in structure and have low computational intensity.

Overall, the TRS technique demands costly and complex instruments along with good contact between the sample and the detector [33]. With the rapid development of optoelectronics, the TRS technique needs to be improved in the direction of smaller, lower-cost portable instruments for real-time detection of fruits and vegetables before and after harvest. Meanwhile, the achievement of noncontact measurement is a difficult problem for the TRS technique.

### 4.2. Spatially Resolved Spectroscopy

The SRS technique acquires reflectance from a point light source of stable intensity at different distances, and the inverse algorithm of the diffuse approximate equation is employed to estimate the $\mu_{a}$ and $\mu_{s}$. Since the path of light transfer is “banana-shape” in turbid media [22], shown in Figure 7a, the spatially resolved spectrum contains information at different depths of samples, leading to suitability for analyzing features of tissues with different depths. There are two main detection forms based on the SRS technique, i.e., contact and noncontact. The contact form relies on a single detection fiber or fiber array, while the noncontact form uses imaging equipment.

Single fiber and multifiber array are two configurations normally used in terms of fiber arrays. As shown in Figure 7a, single fiber configuration is flexible to change the detection position and distance by moving the detection fiber. However, the movement of the detection fiber may cause measurement errors and be time-consuming, owing to different surface curvatures for samples. Multifiber array configuration is capable of solving the problem of the single fiber needing to move the detection fiber frequently in the measurement. Zhou et al. [61] lined up the light-source fiber and multiple detection fibers to fix them in the multiplexer, shown in Figure 7b. However, the spectrum data still require consecutive reading due to the fiber configuration coupled with the multiplexer. With regard to this problem, researchers designed a new configuration to solve it, shown in Figure 7c. The light source fiber and detection fibers have circular distribution and are coupled to the imaging spectrometer of a CCD camera [62], which could simultaneously collect spectral signals at different SDSs. However, this configuration is unsuitable for assessing the samples with curved or irregular surfaces due to the limitations of SDS, the shape and size of rigid probes [63]. On this basis, another SRS system based on a multichannel hyperspectral imager has been developed [39,64]. As shown in Figure 7d, this system is capable of obtaining 30 groups of spectra simultaneously with the SDS from 0.5 to 36 mm. Moreover, this system has good contact between the probe and samples due to the flexible probe. Furthermore, the system enables good optical properties to evaluate the tissues of samples at different depths.

The noncontact SRS system uses a CCD imaging device to capture diffuse images. At present, the noncontact SRS system based on hyperspectral imaging is widely used for quality evaluation of fruits and vegetables. As illustrated in Figure 8a, the system is composed of a CCD camera, an imaging spectrometer and a point light source. Spectral reflectance can be obtained at a certain SDS by using line-scanning hyperspectral imaging to realize noncontact measurement. Numerous pieces of literature have demonstrated that the SRS technique based on hyperspectral imaging was able to measure optical properties for fruit and vegetables effectively and rapidly in the region of 400~1000 nm [25,31,65,66]. For example, Qin and Lu [31] measured optical properties of apples through hyperspectral imaging-based spatially resolved spectroscopy. Combined with MC simulation, the range of penetration depth of light transmission in apple tissues as well as the optical induction place for apples was determined.

Compared with the TRS technique, the SRS technique could measure optical properties effectively over a wider spectral range. In addition, the cost for an instrument based on the SRS technique is lower than that of TRS. However, TRS enables penetration of tissues as deep as about 1 to 2 cm, while SRS usually involves surface information of samples, and thus SRS may be insufficient for analyzing internal defects at deep depths of samples [56]. Although the capability of SRS in penetration depth has been enhanced in later studies [67], it still cannot match TRS in penetration depth. In addition, obtaining accurate information of optical properties in tissues is still challenging for both SRS and TRS techniques.

### 4.3. Frequency Domain Technique

The frequency domain technique normally uses a frequency-modulated beam as an incident light source to obtain the amplitude variation and phase delay angle at a certain radial distance. Combined with the analytical solution of the diffuse approximation equation, the $\mu_{a}$ and $\mu {’}_{s}$ of an object could be estimated [45]. Incident light and emergent light have different amplitudes and phases due to light absorption and scattering, which is the principle of this technique. Cletus et al. [68] measured the $\mu_{a}$ and $\mu {’}_{s}$ within 710~850 nm based on a tunable laser employed by a wideband tunable instrument, proving the wide applicability of tunable frequency instruments. Until now, the FD technique combined with near-infrared spectroscopy is widely applied to the medical field [69], while there is little research in the field of agricultural engineering.

Both FD and TRS techniques require good contact between samples and the detector with similar arrangements in light source and probe. Different from the pulse laser of the TRS technique, the light source of the FD technique is a continuous laser source after frequency modulation, and optical properties are measured at a single frequency. Additionally, the instrument based on the FD technique is relatively simple and cheap.

### 4.4. Structured Illumination

Recently, structured illumination (SI) has been used for optical imaging of biological tissues. Different from the single-point light source of the TRS, SRS, and FD techniques, the SI technique provides 3D profilometry and depth-resolved features and enables an upgrade of spatial resolution and contrast, leading to better control of depth sensitivity by adjusting spatial frequency. Based on the purpose of the applications, the SI technique can be realized by inverse or forward methods. The two methods mainly differ in the method of image processing, called spatial frequency domain imaging and structured illumination reflectance imaging [70,71].

#### 4.4.1. Spatial Frequency Domain Imaging

Spatial frequency domain imaging (SFDI) based on the inverse method is a noncontact widefield optical imaging technique, which can visualize optical properties in the spatial frequency domain by spatially modulated illumination. Based on diffusion theory, the spatial modulation source is introduced into a steady-state diffusion equation to realize spatial resolution measurement. In 1998, the SFDI method was first proposed to measure optical properties of biological tissues and was further used for the measurement of optical properties [72]. Subsequently, the SFDI technique was applied in the field of agricultural engineering for the first time in 2007 [70] to achieve the detection of apple damages. Hu et al. [73] applied a low-cost SFDI system to further evaluate the optical properties of internal browning in apple tissue. Since then, more and more studies on SFDI technique have been used to evaluate the optical properties and quality of fruits and vegetables such as pears [74,75] and peaches [76].

As shown in Figure 8b, the SFDI system mainly contains a light source, a projection device and an acquisition device. During the process of measuring optical properties, reflectance images are obtained from samples with various spatial frequencies of illumination, and then the direct component (DC) and alternating component (AC) images are obtained by image demodulation. Finally, the $\mu_{a}$ and $\mu {’}_{s}$ are estimated by the analytical diffusion equation. Among them, image demodulation is the key step, and researchers can choose the frequency and number of phases according to the actual situation. Different demodulation methods affect the efficiency of SFDI, and there are three commonly used demodulation methods, i.e., three phase demodulation (TPD), single snapshot demodulation (SSD) and spiral phase demodulation (SPD). The TPD method proposed by Cuccia et al. [77] provides a good plan for diffuse reflection image acquisition, but it requires three different phase offset images to calculate reflectance, leading to a long image acquisition time. In view of the shortcomings of the TPD method, Vervandier and Gioux [78] proposed the SSD method, which only requires one reflected image so as to improve the acquisition efficiency. However, when this method divides the spatial spectrum into the DC and AC spectra, information will be lost, yielding the images with lower quality. For solving this problem, Nadeau et al. [79] proposed the SPD method, which applied the spiral phase function in a two-dimensional Fourier space to carry out two-dimensional Hilbert transformation, in which the spiral function was used to get spatial frequency information from the sinusoidal mode of rotation, and the speed of data acquisition is two to three times greater than that of the TPD method. Since then, researchers have begun to carry out many innovations in demodulation methods to further enhance efficiency [75,80].

The SFDI is capable of mapping the $\mu_{a}$ and $\mu {’}_{s}$ in pixel units, and the two-dimensional optical properties distribution are obtained by one measurement. In addition, since the light attenuation in tissues depends on spatial frequency, light penetration depth is also related to spatial frequency. The SFDI technique also provides depth resolution information about structure and composition of tissues through changing the modulation frequency of illumination. Based on this, Hayakawa et al. [81] reported quantitative metrics for optical sampling depth in the spatial frequency domain by the MC method and provided a way to establish optical sampling depth for arbitrary optical properties.

#### 4.4.2. Structured Illumination Reflectance Imaging

The structured illumination reflectance imaging (SIRI) technique based on the forward method is a new means for quality assessment of horticultural or agricultural products and analyzes the DC and AC images by illuminating objects with a sinusoidal pattern of a single spatial frequency. Light penetration depth could be controlled by varying spatial frequency, and thus this technique is beneficial to detect subsurface defects [71]. Lu et al. [21] applied the SIRI technique to detect bruised apples for the first time. As shown in Figure 8b, the optical configuration for the SIRI technique is similar to that for the SFDI technique, and the image can be obtained through the same demodulation method. Different from the SFDI technique based on an inverse algorithm to estimate optical properties, the SIRI technique directly analyzes the DC and AC images to realize the defect detection, which saves time. However, the SIRI technique requires different spatial frequencies when analyzing biological tissues at different depths, yielding longer times for image acquisition due to the single-frequency sinusoidal mode. To solve this problem, Lu and Lu [82] used composite sinusoidal patterns in SIRI to effectively enhance the performance of apple defects detection.

By utilizing sinusoidal modulated structural-light patterns, SIRI provides two separate sets of images (the DC image and the AC image). The DC image corresponds to uniform illumination, while the AC image results from structured illumination. Compared with the DC image, the AC image has better spatial resolution and image contrast and a depth-resolved ability. Based on this, Li et al. [83] detected apple bruises at different degrees in terms of AC and DC images. The results discovered the superiority of AC images in measuring subsurface defects. Due to different defect characteristics provided by the DC and AC images, the combination of them may improve detection accuracy. Lu and Lu [84] developed a multispectral SIRI system based on liquid crystal tunable filters (LCTF). The results showed that the ratio image is equal to or better than the DC and AC images for the detection of apple bruises. Thereafter, Sun et al. [71] used a multispectral SIRI system based on DC, AC and ratio images coupled with the image classification method to detect and classify early diseases of peaches.

Similar to the SFDI technique, the SIRI technique obtains images of tissue based on spatial frequency and light penetration depth. By determining light penetration to the location of damaged tissue, it is possible to reveal subsurface tissue properties at specific depths and enhance the visibility of defects, but SIRI is still not suitable for online inspection of food quality attributes due to time consumption. Up to now, the SIRI technique has only been employed in defect detection for some fruits and vegetables, i.e., apples [21,84,85], oranges [86] and cucumbers [87], which means that the SIRI technique needs to be further developed to meet the demands of quality and safety detection for fruit and vegetables.

## 5. Applications

As mentioned above, light penetration depth in fruits and vegetables can be acquired by the $\mu_{a}$ and $\mu {’}_{s}$, which were obtained from the TRS, SRS, SFDI and SIRI techniques combined with each corresponding model or algorithm. Meanwhile, the maturity, quality evaluation and defect detection could be predicted by optical parameters. This section will focus on the application of optical properties to quality assessment in fruits and vegetables.

### 5.1. Analysis of Absorption and Scattering Coefficients

The absorption and reduced scattering coefficient spectra in different wavelengths and conditions can be given by experimental measurements. Many studies have reported that different fruits and vegetables present different patterns of spectra, especially for some specific bands, such as 670 nm and 970 nm [65,88,89]. Moreover, with the increase in wavelength, the $\mu_{a}$ spectrum fluctuates, while the $\mu {’}_{s}$ decreases.

The studies of optical properties measurement for quality evaluation of fruits and vegetables are presented in Table 3. Sun et al. [34] researched optical properties of peaches in fungal infection and classified peaches according to $\mu {’}_{s}$ and $\mu {’}_{s}$. In the absorption coefficient spectrum, distinct absorption peaks appear at 675 and 970 nm, individually, due to chlorophyll in tissues and water absorption. Huang et al. [90] collected spectra of peaches in the range of 550~1650 nm and found that absorption peaks also appear at 750 nm, 1180 nm and greater than 1400 nm due to water absorption and the combination of C–H, N–H and O–H bonds in tissues. In addition, absorption peaks may also occur at 550~600 nm due to the presence of anthocyanins. A similar trend is also seen in the absorption spectra of other fruits, including apples [63], kiwifruits [91] and pears [53]. The above studies proved that the absorption spectrum has high sensitivity at different wavelength points and is suitable for the detection of maturity and quality, while the curve is relatively flat in the spectrum of $\mu {’}_{s}$ and gradually decreases with the increase in wavelength, which is consistent with the conclusion of the theory of Mie scattering. The $\mu {’}_{s}$ is related to the physical properties of tissues such as cell size, compact degree and fine structure. As the peaches matured from hard to soft, the sharp decline of $\mu {’}_{s}$ could not meet the conditions of a strong scattering medium, leading to the invalidation of the diffusion model [88]. Yang et al. [55] measured the optical properties at different tissue layers (exocarp, green mesocarp and pulp) among different positions (stem, equator and calyx) of “Huanghemi” melons in the spectral ranges of 450~850 nm and 950~1650 nm, with the light penetration depth at the corresponding positions calculated, and found that the $\mu {’}_{s}$ values all decreased with the increasing wavelength, but the $\mu {’}_{s}$ values were varied due to differences in cell structure, which indicated that the scattered particles in the sample have different densities and sizes [92]. Therefore, the $\mu {’}_{s}$ could be applied to evaluate the firmness of fruits and vegetables. In terms of bruise detection, Zhang et al. [93] analyzed optical properties of healthy and bruised blueberries. The results found the $\mu {’}_{s}$ varied significantly between two kinds of blueberries, which proved that the $\mu {’}_{s}$ could be used to identify healthy and damaged blueberry tissues. Gao et al. [54] also reported the $\mu_{a}$ and $\mu {’}_{s}$ of kiwifruits at 950~1650 nm, finding that $\mu {’}_{s}$ first decreased and then increased for bruised samples, while $\mu_{a}$ had no significant difference.

As shown in Table 3, in addition to the IS (including SFDI and SIRI) technique, most studies have focused on SRS due to low cost and high accuracy, followed by TRS, while SFDI and SIRI techniques began to be widely used after the year 2016. Since absorption and scattering properties are related to chemical composition and physical properties of biological tissues, respectively, the light absorption coefficient fluctuates significantly in the carotenoids (about 500 nm), anthocyanins (550~600 nm), chlorophyll (mainly chlorophyll a at 678 nm) and O–H bond (970 nm and 1400 nm). Thus, quality parameters such as firmness, SSC, maturity and defects can be effectively assessed by $\mu_{a}$ and $\mu {’}_{s}$ spectra, even spectral combination, among which the $\mu_{a}\times \mu {’}_{s}$ and $\mu_{eff}$ have more comprehensive information. It is worth noting that the $\mu {’}_{s}$ is greater than $\mu_{a}$ for most fruits and vegetables, indicating that they are scattering dominant tissues and meet the conditions of the diffusion equation ($\mu {’}_{s}$ >> $\mu_{a}$).

### 5.2. Maturity and Quality Assessment

Maturity is one of the important indexes to determine the picking time of fruits and vegetables and to evaluate quality after harvest. Maturity is generally highly relevant to several quality parameters, such as firmness, SSC, pH, starch content and the color of pericarp. The growth of fruits and vegetables is often accompanied by changes in chemical composition and physical structure. Since the $\mu_{a}$ and $\mu {’}_{s}$ are related to the chemical composition and physical properties, respectively, they can be used to evaluate the maturity of samples. Qin and Lu [98] measured the ratio of $\mu_{a}$ at 675 nm and 535 nm for tomatoes at three ripeness stages to compare the optical properties at different maturities of tomatoes. The absorption of tomatoes changed significantly at different maturity stages, while the $\mu {’}_{s}$ decreased with the tomato ripeness on account of the changes in anthocyanin and chlorophyll contents in the tissues. Huang et al. [89] extracted $\mu_{a}$ and $\mu {’}_{s}$ from spatially resolved diffuse reflectance within 550~1300 nm to assess tomato maturity. The classification accuracies of tomatoes at six maturity stages were compared with those at three ripeness stages by using $\mu_{a}$, $\mu {’}_{s}$ and their combinations. The results uncovered that the overall recognition rate was higher when tomatoes were divided into three ripeness stages with classification accuracy up to 94%. Sun et al. [99] divided 330 “Red Star” peaches into three ripeness stages (relatively immature, commercially ripe and overly ripe) according to firmness and analyzed the optical coefficients of bruised peaches at different ripeness stages, with the highest accuracy rates up to 94.12%, 95.59% and 94.12%, respectively. The results showed that both the changes in maturity and sample defects would result in a reduction in $\mu {’}_{s}$ and an increase in $\mu_{a}$, and the optical properties of immature peaches changed more after bruising than others. The absorption coefficients changed significantly due to the dual effects of maturity and bruising, which means that $\mu_{a}$ is more sensitive to changes in wavelengths and has the potential to better evaluate the optical properties of peaches compared with $\mu {’}_{s}$. Table 4 shows the recent studies applying the TRS, SRS and SFDI techniques to assess maturity and quality attributes of various objects.

As shown in Table 4, firmness and SSC are considered to be internal characteristics significantly related to maturity, and thus most studies focus on the prediction of them. Cen et al. [65] applied hyperspectral imaging-based spatially resolved spectroscopy to obtain $\mu_{a}$ and $\mu {’}_{s}$ of peaches at 515~1000 nm. The prediction models for peach quality were established by using $\mu_{a}$ and $\mu {’}_{s}$ spectra and their combinations. They found that $\mu_{a}$ and $\mu {’}_{s}$ correlated with maturity/quality attributes, and the correlation coefficients were 0.42~0.855 and 0.204~0.84, respectively. Similar research in the application of kiwis and apples has also been reported by Ma et al. [63,100] and Huang et al. [101] (tomato). In addition, compared with individual $\mu_{a}$ and $\mu {’}_{s}$, the combined spectrum of $\mu_{a}$ and $\mu {’}_{s}$ could generally improve prediction results. In order to further enhance the accuracy of firmness and SSC prediction, Huang et al. [90] proposed a spectral difference method to calculate the spectral difference between the first SDS spectra and the remaining 14 SDSs’ spectra based on the SRS technique, obtaining difference reflectance (DR) spectra. They found that the model using the DR spectra could more effectively improve predicted results and reduce the influence of the shallow layer on quality prediction, the best correlation, up to 0.853 and 0.839, separately.

**Table 4 foods-13-02688-t004:** Maturity and quality parameters of various fruits and vegetables evaluated by TRS, SRS, FD and SFDI techniques in recent years.

Product	Technique	Year	Parameter	Wavelength (nm)	Model	Result	Reference
Apple	TRS	2020	Maturity	540~1064	Pearson correlation analysis	For chroma: r$\mu_{a}$ = −0.604, r$\mu {’}_{s}$ = 0.615	[102]
	SRS	2020	Maturity	550~1650	PLSDA	GD: Acc = 100%	[103]
	SRS	2021	Firmness, SSC	600~1000	PLSDA	SSC: R^2^ = 0.92, Rmse = 0.35% Firmness: R^2^ = 0.87, Rmse = 0.71 N	[63]
	SRS	2021	Flesh color	190~1070	PLSDA	Skin: R^2^ = 0.95 Whole flesh, R^2^ = 0.69	[104]
	SFDI	2022	SSC, firmness, color	450~750	SVM	R_p_ = 0.66, 0.73, 0.86	[105]
Peach	SRS	2020	Soluble, TA	550~1000	Pearson correlation analysis	r = 0.898 r = −0.776	[97]
	SRS	2021	Maturity	550~1000	SVM	Acc = 94.12%, 95.59% and 94.12%	[99]
	SRS	2022	Firmness, SSC	550~1650	PLSDA	R_p_ = 0.853, Rmse = 14.76 N R_p_ = 0.839, Rmse = 0.5	[90]
Kiwifruit	SRS	2022	Firmness, SSC, pH	650~1000	PLSDA	R^2^ = 0.64, 0.67, 0.38 Rmse = 3.63 N/cm^2^, 0.95%, 0.19%	[100]
Mango	TRS	2015	Maturity	540~900	EP, FD	——	[106]
	TRS	2023	Maturity, pulp color	540	Logistic/exponential model	$R_{\mathrm{adj}}^{2}$ = 99.8%	[57]
Tomato	SRS	2018	SSC, pH	400~1100 900~1300	PLSDA	For 400~1100: R_p_ = 0.729, 0.743 For 900~1300: R_p_ = 0.815, 0.741	[101]
	SRS	2018	Firmness	550~1650	PLSDA	R_p_ = 0.859, Rmse = 1.00	[66]
	SRS	2020	Maturity	550~1650	SVM	Acc = 98.3%	[67]

r = Pearson correlation coefficient, PLSDA = partial least square discriminant analysis, GD = golden apple, SVM = support vector machine, Acc = accuracy, R_p_ = correlation coefficient, TA= titratable acid, R^2^ = coefficient of determination, EP = ethylene production model, FD = firmness decay model, R_adj_^2^ = r-squared adjusted.

The ripeness process of samples after harvest normally comes with changes in cellular structure and optical properties. In the study on the “Baifeng” and “Xiahui 8” peaches, Ma et al. [52] reported that firmness, intercellular space rate and cell wall thickness were well correlated with optical scattering. Zerbini et al. [106] also built a model of mangos and nectarines at different ripeness stages by measuring $\mu_{a}$ using time-resolved spectroscopy, providing an assessment of the ripeness index for individual fruits.

For the maturity and quality assessment of fruits and vegetables, the studies summarized in Table 4 mainly focused on prediction and classification by mathematical models. The results were normally described by R^2^ or R_p_ and Acc, respectively. Rmse was also used as evaluation indicators for prediction results. The higher R^2^ or R_p_ and Acc with the smaller rmse refer to the better results, which demonstrates that the method used for evaluating the quality of fruits and vegetables is superior. Most of the studies in Table 4 acquired SRS and TRS techniques to predict maturity, firmness, SSC, pulp color and acidity, which indicate that SRS and TRS are more suitable for maturity and quality assessment of fruits and vegetables than the SI technique. Besides, both $\mu_{a}$ and $\mu {’}_{s}$ were related to firmness, SSC, pulp color and starch index. However, for a few very soft fruits, the $\mu {’}_{s}$ pattern might change, which means that the assumptions of the diffusion model would be not valid and would result in large errors, which needs to be further studied. However, as fruits ripen, the sharp decline in $\mu {’}_{s}$ cannot satisfy the hypothesis of the diffusion model, leading to obtaining erroneous $\mu_{a}$ and $\mu {’}_{s}$ spectra, which needs further study.

### 5.3. Defect Detection

During harvesting, fruits and vegetables are subject to various types of mechanical damages, such as punctures, cracks, abrasions and bruising, which could make them susceptible to fermentation, rot or mold through storage and even infect other healthy products. These damaged fruits and vegetables easily affect the purchasing behavior of consumers and cause significant economic losses to the market. Therefore, it is necessary for fruits and vegetables with surface defects or internal defects to be separated from healthy ones. At present, the TRS, SRS, SFDI and SIRI techniques have been applied for defect detection.

Table 5 lists recent studies using these optical techniques combined with models, such as convolutional neural networks and PLSA [107,108,109,110], to complete defect detection of fruits and vegetables. Sun et al. [99] obtained optical properties of damaged peaches at different maturities using the spatial resolution method. Based on this, a support vector machine models by using $\mu_{a}$ and $\mu {’}_{s}$ spectra were established for identifying defective peaches, and the accuracy for good samples was up to 100%, which indicated that optical properties are capable of classifying healthy and defective products and identifying earlier bruised tissues. Vanoli et al. [94] established classification models based on absorption and scattering properties using the TRS technique to detect internal browning (IB) in apples. They found that TRS could be affected by the number of measuring points, as with the increased number of measuring points, the optimal classification accuracies of healthy and IB fruits were 90% and 71%, respectively. The report showed that the TRS technique could innocuously ascertain the internal browning of intact samples, as the $\mu_{a}$ and $\mu {’}_{s}$ obtained at 780 nm changed noticeably with increased browning. Lu et al. [111] applied the SRS technique to evaluate optical properties of apples, finding that the values of $\mu {’}_{s}$ for healthy fruits were higher than those of bruised fruits, and the value decreased as time increased at the same wavelength, which demonstrated that SRS coupled with the $\mu {’}_{s}$ has potential in defect detection. He et al. [74] used the corrected SFDI system for defect detection of “crown” pears. By comparing the values of $\mu {’}_{s}$, healthy pears and bruised pears could be well distinguished, with classification accuracies of bruised and healthy pears of 98.33% and 100%. Luo et al. [19] also distinguished the quadruple classification of pears and three-way classification, with the accuracy rates at 675 nm of 83.8% and 93.8% by the linear discriminant analysis model established based on the measured $\mu_{a}$ and $\mu {’}_{s}$, respectively. The report demonstrated the potential of the SFDI technique to detect various damage types in fruits and vegetables. In addition, for identifying invisible early bruising, Sun et al. [112] proposed new methods, three-phase coupled curve fitting (TP-CF) and single snapshot coupled lookup table (SS-LUT), and they were used for optical property mapping of non-bruised and early-stage bruised apples, and the reduced scattering coefficient mapping was able to detect early-stage bruised apples. Subsequently, Yu et al. [113] proposed a profile-based correction method for the diffuse reflectance after demodulation in SFDI, which improved the accuracy of optical property measurement for spherical fruit (the standard deviations of $\mu_{a}$ and $\mu {’}_{s}$ were reduced by about 70% and 40%, respectively) and enhanced the detection accuracy of early-stage bruised apples through improved optical property mapping.

Different from other techniques, the AC image in the SIRI technique is able to enhance image resolution and contrast, revealing more certain features or properties compared with the DC image [84], which is better to improve the identification accuracy of defects. Sun et al. [92] selected three characteristic bands of 781, 824 and 867 nm in the structured spectra and applied the WSA for bruise detection of peaches at three maturity levels (S1~S3). The results found that the WSA could solve various image segmentation problems and is suitable for images with high contrast. Moreover, among the three maturity levels, the detection rates of AC images were better than those of DC images, with recognition rates up to 92%, 97.43% and 99.86%, respectively, which proved the superiority of AC images and showed that S-MSI, coupled with a proper image-detection method, could improve the detection performance of early-stage bruised peaches. Cai et al. [86] extracted the features of DC, AC and ratio images by the SIRI technique and established four classification models (i.e., PLSDA, SVM, least squares-support vector machine (LS-SVM) and k-nearest neighbor (KNN)) to identify infected oranges. The results showed that the classification for DC and AC images coupled with the LS-SVM model obtained better results with accuracies of 88.6% and 92.1%, respectively, compared with other models, while the ratio images combined with the PLSDA model could achieve the optimal classification accuracy of 97.1%, which proved that the SIRI technique is a potential technique to identify early decay in fruits and vegetables. In addition, Lu et al. [87] first reported the use of the SIRI technique to detect subsurface bruising of pickling cucumbers, with an averaged accuracy of 94%.

As shown in Table 5, compared with TRS and SRS, the SI (including SFDI and SIRI) technique gives the highest accuracy in the defect detection of fruits and vegetables, especially in early-stage bruise identification, due to the benefit of noncontact wide field and varying depth resolution, resulting in a recent increase in research on early-stage bruising detection of fruits and vegetables by using the SIRI technique. On the other hand, the appropriate mathematical models also enhance the accuracy of defect detection. With the development of deep learning, the traditional models such as PLSDA, SVM and LDA in defect detection have been gradually replaced. In addition, more and more researchers have begun to seek new models of deep learning to improve accuracy. Hu et al. [117] applied the SVM and CNN models to distinguish two-category (intact and bruised) and three-category apples (intact, mildly bruised and severely bruised) based on the SFDI technique. The results found that CNN classification was superior to SVM, and the recognition accuracy of two and three types of apples was 99.16% and 91.50%, respectively, proving the great potential of deep learning in the improvement of classification accuracy. Additionally, although the above studies demonstrated that optical properties are capable of detecting defects in fruit and vegetable tissues, there are still some practical problems in the application of TRS, SRS, SFDI and SIRI techniques in real-time detection of internal defects in tissues, as described in Section 4.

### 5.4. Analysis of Light Penetration Depth

Light penetration depth could commonly depict the efficient scope or susceptive area of light interaction with object tissues. The photon transmission path in biological tissues can directly show the light penetration depth, such as the “banana shape” path in the SRS technique. Meanwhile, light transmission in fruit and vegetable tissues is a complex process involving photon absorption and scattering. In addition, absorption and scattering properties are related to chemical composition and physical properties, respectively, which could affect the transmission law of photons in biological tissues so as to change light penetration depth. For the samples with thicker skin, such as citrus with rind thickness of 3 to 5 mm, light penetration depth may be affected due to changes in optical properties resulting from the different compositions of pericarp and flesh. Light penetration depth can not only directly reflect the attenuation of light in tissues but also help to obtain longitudinal information of different hierarchical structures [30], which encourages many researchers to study light penetration depth.

Back in 1985, Wilson et al. [119] studied the attenuation of light within 400~800 nm in mammals in the biomedical field, and they found that the effective penetration depth is a complex function of absorption and scattering. Del Bianco et al. [120] investigated the penetration depth of photons in a semi-infinite uniform medium in time domain and continuous waves cases. They found that for time resolution measurement, penetration depth increases with increments of time-of-flight, while for continuous wave measurement, penetration depth increases as the SDS increases. Sarkar et al. [60] applied time-of-flight to measure the penetration depth based on time-resolved reflection spectra, also finding the same conclusion about the relationship between light penetration depth and SDS. Under certain conditions, a penetration depth formula could be derived based on the diffusion equation, such as Equations (6) and (7). On this basis, combined with TRS, SRS, FD or SFDI techniques, $\mu_{a}$ and $\mu {’}_{s}$ could be obtained to calculate penetration depth. Qin and Lu [31] got the penetration depth of 600 “golden delicious” apples in the range of 0.43~8.67 cm through the $\mu_{a}$ and $\mu {’}_{s}$ based on spatially resolved hyperspectral imaging, which is similar to what Fraser [29] reported. Hu et al. [24] applied the SFDI technique to realize the changes of penetration depth in apple tissues by adjusting the spatial frequency of illumination, and the optimal combination of start and end frequencies was analyzed. Low frequency illumination could penetrate deeper into tissue. They further studied several sets of simulation results for typical $\mu_{a}$ and $\mu {’}_{s}$ to obtain an image of the relationship between light penetration depth and spatial frequency [121].

Another method called Monte Carlo could be used to determine light penetration depth, because it is capable of describing the photon transmission law flexibly and accurately, and it was also applied in biological tissues that don’t need to meet the assumptions of the diffusion model. Sujatha et al. [26] utilized the MC method to simulate multilayer skin tissues, and the optimal SDS was determined by obtained penetration depth information. The study reported that average penetration depth increased with the increment of SDS and gradually got saturation to a value close to the actual depth of each layer of tissue. A similar conclusion was found in Wang’s studies [25]. In the field of agricultural engineering, to assess the influence of fruit cores and pericarp on photon transmission, Ding et al. [122] took peaches as research objects to simulate the photon transmission process in the pericarp, flesh and core successively. They found that since the photons were absorbed/scattered in tissues during transmission, light penetration depth began to decrease, and the light intensity reaching the boundary of the fruit core was small, which suggests a small influence of fruit cores on optical transmission characteristics. When the pericarp thickness increases, the detection efficiency will decrease, especially near the incident point of the photon. Therefore, the influence of pericarp on light transport features is greater than that of the core. For the flesh layer, tissues with smaller $\mu {’}_{s}$ allow light to penetrate deeper. In addition, the selection of SDS, detection angle and light source intensity also affect the detection efficacity and sensitivity. Deeper information in fruit and vegetable tissues could be obtained by increasing the SDS. Xu et al. [123] detected light penetration depth in “Hami” melon tissues in the spectral range of 720~880 nm. For penetration depths of different SDSs (10, 20 and 30 mm), the depth values were 12.3, 14.9 and 19.0 mm, respectively, with single wavelengths, which indicated that SDS is an important parameter for the design of an optical probe to ensure the efficient assessment of internal quality for fruits and vegetables. On the other hand, Shi et al. [30] took thin-skinned apples and thick-skinned oranges as detection objects by using the MC method to analyze the influence of pericarp thickness on light penetration depth. They found that the pericarp of such thin-skinned fruits as apples had little influence on penetration depth, while like in citrus fruit, light penetration depth decreased with the increased thickness of fruit pericarp. Similar results were also found in melons [55], and they also indicated that the light penetration depth was affected by the difference in SDS and changes of peel thickness.

Table 6 shows that the light penetration depth is affected by several factors, such as wavelength, frequency, SDS and peel thickness, etc., which makes it lack a unified calculation standard. The depth of light penetration is related to the attenuation of light in fruits and vegetables, that is to say, absorption and scattering properties and the determination of the depth of light penetration help to evaluate the optical properties of multilayer tissues. In addition, the above studies indicated that the analysis of light penetration depth is helpful in detecting the optical path layout and detection scheme of a system, so as to further improve the ability to assess the quality of fruits and vegetables.

## 6. Challenges and Prospects

In recent years, optical detection techniques such as TRS, SRS, FD and SI have been rapidly developed. Absorption and scattering properties could be obtained with appropriate techniques, which shows the potential to assess the quality of fruits and vegetables more comprehensively and also provides a new means for the quality evaluation of the same. Therefore, more and more attention has been paid to the explorations of optical properties of fruits and vegetables. In the early stage, researchers usually regarded fruits and vegetables as a single uniform structure in order to reduce the calculation of parameter variables due to the immaturity of the techniques. In later research, they found that the assumption of a single layer would introduce errors into measurements, since most fruits and vegetables are composed of pericarp and pulp or even multilayers of heterogeneous tissue [25]. Subsequently, studies focused on the optical properties for samples with two or even multilayers of tissue have begun to be developed, especially related to the light penetration depth, since it is helpful to explore the tissue information of different layers and to mine the comprehensive information of objects to achieve the improvement of detection accuracy. The light penetration depth is capable of determining whether the photons reach the pericarp, flesh or core to confirm obtained information in corresponding tissue layers. Therefore, studies on light penetration depth are necessary and significant because it contributes to designing more effective detection devices for biological tissues with multilayers, i.e., citrus, tomato and watermelon [30], and also to providing a theoretical basis for subsequent research on the multilayer structure information of samples. However, there are still some problems in the current research.

Firstly, there are no unified standards for the penetration depth. Wilson and Jacques [27] and Fraser [29] had different views about defining light penetration depth with luminous intensity, respectively. The former defined it as the propagation distance in which luminous intensity is reduced by 1/e (about 37%), while the latter regarded it as the distance where luminous intensity is down to 1%. The difference between the two is clearly seen in Figure 3. In terms of theoretical models, under the approximation condition of steady-state diffusion, the value of light penetration depth can be calculated according to Equations (6) and (7), respectively, while the MC method describes light penetration depth in biological tissues by determining the maximum longitudinal position that photons reach in tissues. The diffusion approximate equation is only applicable to the strongly scattering medium, while the MC method is more flexible and accurate, which contributes to the MC method being more widely used for light penetration depth. However, the MC method requires a lot of computation and more time in the simulation process. In order to accelerate the simulation of MC methods, new methods such as scaling MC, perturbation MC, hybrid MC, etc. have been developed [38], which need to be further applied to research of light penetration depth to verify their feasibility.

Secondly, the determination of light penetration depth is affected by many factors. In the TRS technique, it is related to the time required for incident light to reach the camera [60], whereas in the SFDI technique using structured illumination, light penetration depth increases with the selected spatial frequency [121]. These studies have also found that light penetration depth correlates with SDS, luminous intensity and angle of incident light [122]. The larger the SDS is, the greater the measured penetration depth is [25], which indicates that it is feasible to measure optical properties of pericarp and flesh tissues accurately by adjusting SDS. The absorption and scattering properties vary at different wavelengths due to complex chemical composition and physical structure in fruit and vegetable tissues, and thus penetration depth is also indirectly related to the wavelengths [31,55]. In the application of apples and melons, researchers have reported on the fluctuation of penetration depth under different wavelengths. The relationship between wavelength and penetration depth is conducive to accurate measurement of specific wavelengths for the object. In addition, thickness of pericarp is also a significant factor that affects light penetration depth. When the above-mentioned aspects are solved or developed, the study of light penetration should be beneficial in designing the most suitable detection device so as to obtain more comprehensive information for fruits and vegetables.

The studies on light penetration depth provide a theoretical basis for evaluating the internal quality of fruits and vegetables based on spectroscopy. Although much literature has reported light transmission properties in fruit and vegetable tissues, there have been few actual studies on penetration depth of fruits and vegetables since it is difficult to measure. With more and more attention being paid to research on light penetration depth on two-layer or even multilayer tissues, it is helpful to understand the attenuation of light and judge the obstructive effect of pericarp, flesh or core on light transmission, ensuring that accurate optical properties information is obtained in each layer of tissue, so as to enhance the accuracy of quality assessment for objects.

## 7. Conclusions

Light penetration depth reflects the attenuation of light in biological tissues and is an optical parameter dependent on absorption and scattering properties. Studies on light penetration depth in fruits and vegetables have been carried out over the past two decades. However, there is no uniform standard for the calculation of light penetration depth due to different definitions (light penetration depth is defined as the distance traveled when the light intensity is reduced to 37% or 1% or the maximum longitudinal position reached by the photon in the tissue) and diverse theoretical models (diffusion equation, MC and AD methods) used by researchers. With the rapid development of emerging techniques such as TRS, SRS and SI (including SFDI and SIRI), as well as the novel design of corresponding system sensor configurations, the light penetration depth in biological tissues will be described more accurately. The TRS technique can penetrate deeper into the tissues, which is important for the detection of internal quality or defects of fruits and vegetables, while the SRS technique is widely used in the quality assessment of fruits and vegetables due to the advantages of simple instruments, low cost and fast measurement speed. As a new technique, the SI (including SFDI and SIRI) technique has unique superiority and application potential in the defect detection of fruits and vegetables due to its characteristics of imaging depth discrimination and effective signal enhancement. In addition, these emerging optical detection techniques combined with diffusion equation and MC methods could determine the light penetration depth through $\mu_{a}$ and $\mu {’}_{s}$, which is of great significance for the study of optical properties of fruits and vegetables (such as tomatoes, citrus, etc.) with two or even multilayers of heterogeneous tissues. However, in the process of measurement, the difference of SDS, intensity of light source and angle of incident light also have certain influences on light penetration depth. More effective detection devices for fruits and vegetables would be designed by addressing the impact of the above factors. Meanwhile, it is necessary to study light penetration depth to improve the detection performance for quality and safety assessment of fruits and vegetables.

## Figures and Tables

**Figure 1 foods-13-02688-f001:**
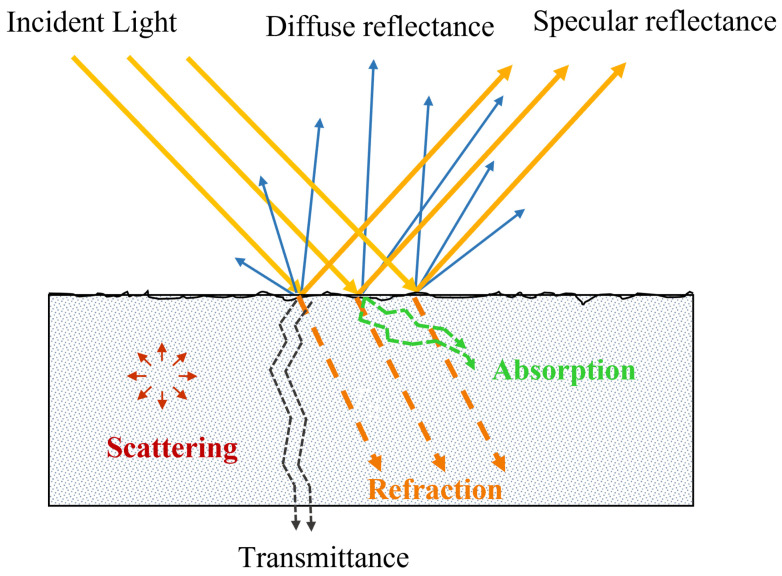
Schematic of the interaction between light and an object.

**Figure 2 foods-13-02688-f002:**
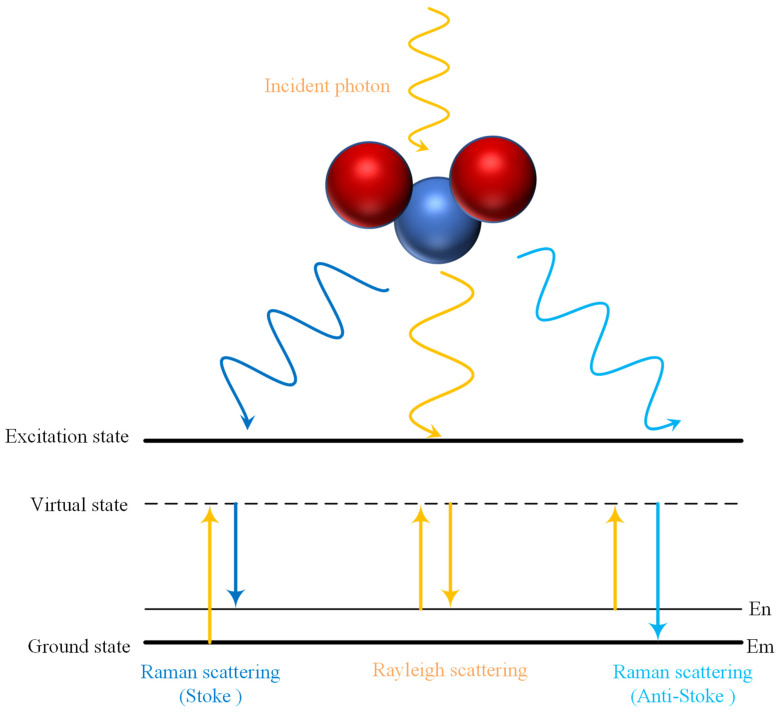
Energy variations of Rayleigh scattering and Raman scattering (energy level: Em < En).

**Figure 3 foods-13-02688-f003:**
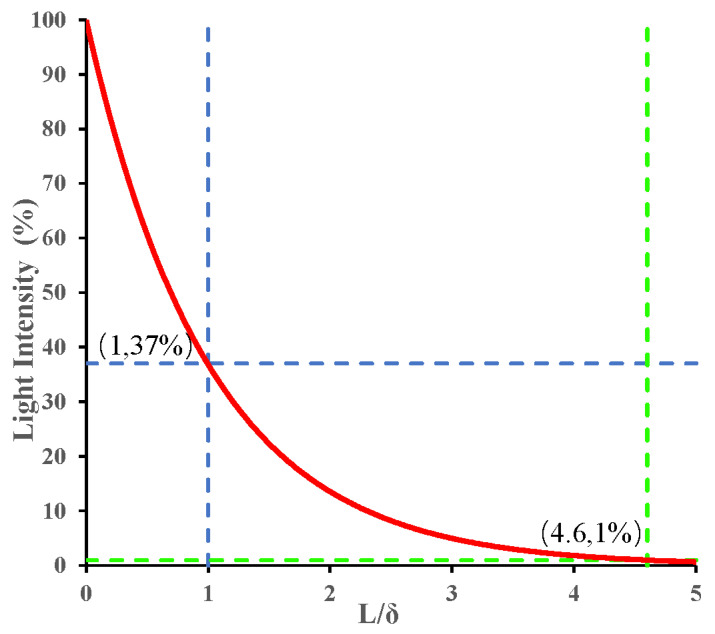
Relationship between light attenuation and light penetration depth in tissues.

**Figure 4 foods-13-02688-f004:**
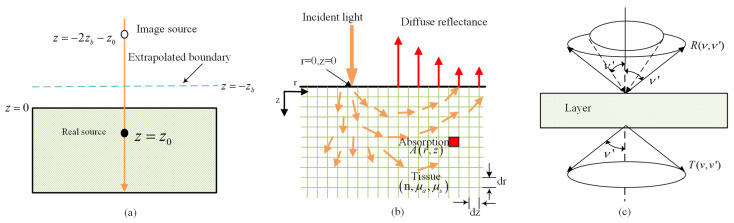
(**a**) Schematic representation of extrapolated boundary; (**b**) MC simulation for diffuse reflectance and absorption of tissues; (**c**) transmission process of photons in the AD method.

**Figure 5 foods-13-02688-f005:**
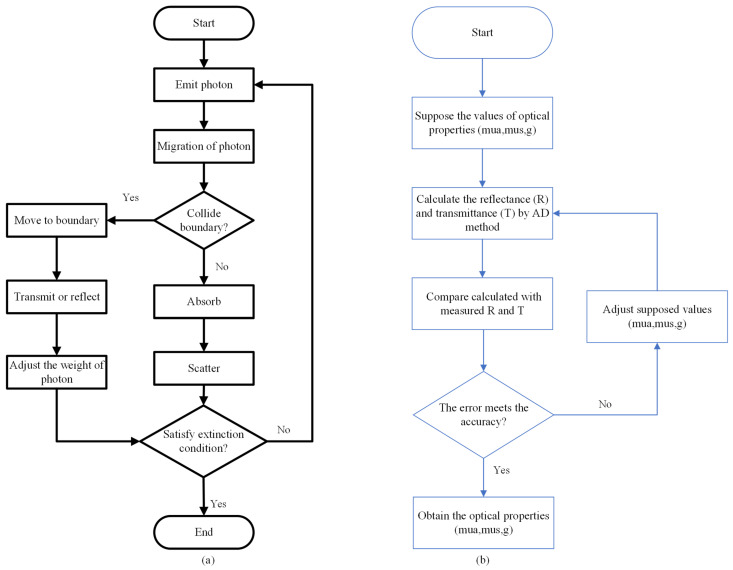
(**a**) Flowchart of the MC simulation of a single photon; (**b**) flowchart of the IAD method.

**Figure 6 foods-13-02688-f006:**
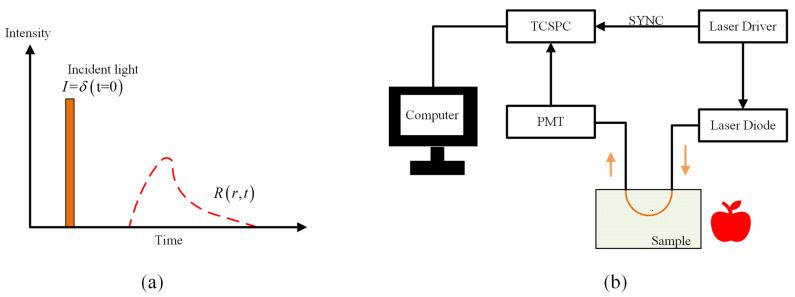
(**a**) Short-pulsed illumination at the surface of a semi-infinite turbid medium; (**b**) schematic of time-resolved system for measuring optical properties, in which PMT is a photomultiplier tube and SYNC is the synchronization signal.

**Figure 7 foods-13-02688-f007:**
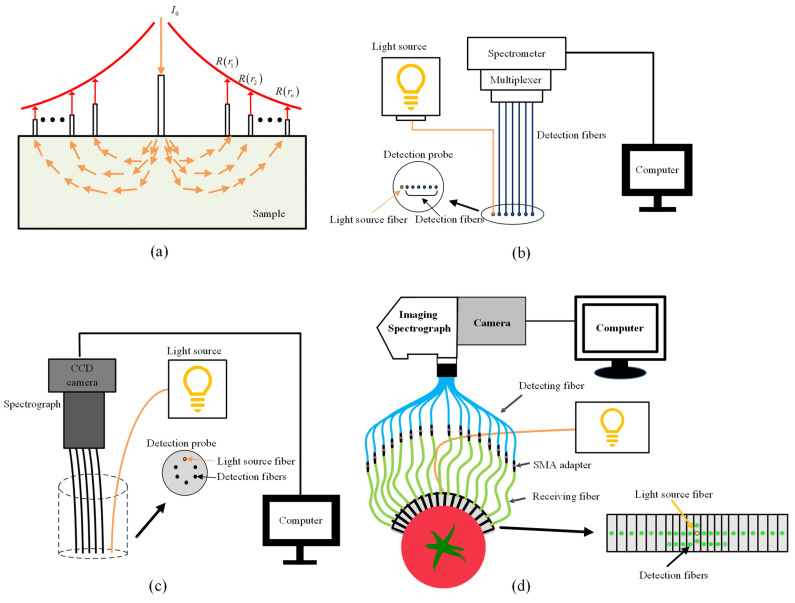
(**a**) Schematic illustrations of configuration of single-fiber and “banana-shape” path of light transfer; (**b**) multifiber array based on a multiplexer; (**c**) multifiber array based on a multiplexer; (**d**) multichannel curved array based on spatially resolved system.

**Figure 8 foods-13-02688-f008:**
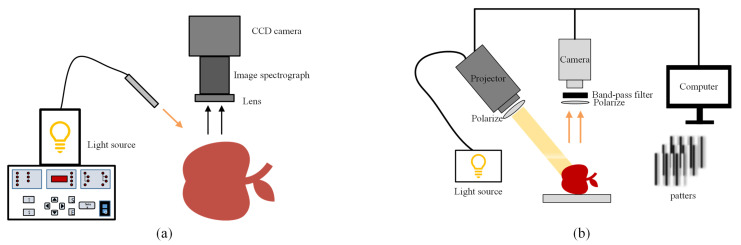
(**a**) Schematic illustrations of noncontact SRS systems; (**b**) schematic of an SFDI system for spectral image acquisition.

**Table 1 foods-13-02688-t001:** Comparison of characteristics of three simplified models.

Model	Advantage	Limitation
Diffusion approximation equation	Computationally fast and intuitive, wide applicability and suitable for strong scattering media	Unsuitable for low scattering media and unable to handle anisotropic scattering
Monte Carlo method	High accuracy, flexibility and wide applicability	Large calculation, time-consuming and difficult to apply in real time
Adding-doubling method	Suitable for anisotropic scattering and multilayer media With high accuracy and short calculation time, suitable for real-time applications	Limited to the assumption of uniform media, only suitable for planar layer media and unsuitable for extremely high scattering

**Table 2 foods-13-02688-t002:** Comparison of characteristics of different optical detection techniques in quality assessment of fruits and vegetables.

Technique	Advantage	Disadvantage
TRS	Good penetration ability, penetrating tissues as deep as about 1 to 2 cm	Requiring expensive and sophisticated instrumentation, as well as good contact between the sample and the detector
SRS	Being cost-effective on instruments, high execution efficiency and support for a wide range of spectral range	Is insufficient for analyzing deep defects inside samples
FD	Having high resolution and strong noise suppression ability, and the instrument used is cheaper than TRS	Requiring good contact between the sample and the detector, only one single frequency
SFDI	Wide field non-contact, imaging depth discrimination and effective signal enhancement	Requiring selection of the feature band and cannot achieve real-time detection
SIRI	Is capable of revealing subsurface tissue properties at specific depths and enhancing the visibility of defects	Being unsuitable for online inspection of food quality and limited to defect detection

**Table 3 foods-13-02688-t003:** Research of optical properties measurement for quality evaluation of fruits and vegetables in recent years.

Product	Technique	Year	Wavelength (nm)	$\mu_{a}$ (cm^−1^)	$\mu {’}_{s}$ (cm^−1^)	Reference
Apple	SRS	2009	500~1000	0.04~2.52	1.02~12.61	[31]
	SRS	2014	500~1000	0~2.6		[62]
	TRS	2014	650~1050	0.02~0.45	——	[94]
	SFDI	2021	400~1000	0~1.2	0~35	[95]
Peach	SRS	2011	515~1000	0~0.5	5~20	[96]
	SRS	2020	600~1000	0.9~4.1	5.59~16.12	[34]
	SRS	2020	550~1000	0~0.5	2~14	[97]
	IS	2020	400~1050	0.3~4	2~12	[52]
	SRS	2022	550~1650	——	——	[90]
Pear	IS	2023	500~1000	0~3	10~18	[53]
Kiwifruit	IS	2021	950~1650	0~9	3~13	[54]
	IS	2022	500~1050	0~0.8	2~10	[91]
Tomato	SRS	2018	550~1300	0~2.5	0.8~6	[64]
Blueberry	IS	2017	925~1400	0~10	0~8	[93]
Mango	TRS	2023	540	0.103~0.403 0.106~0.511	——	[57]
Melon	IS	2024	450~1650	0~7.5	5~55	[55]

IS = integrating sphere technique.

**Table 5 foods-13-02688-t005:** Recent applications of these optical detection techniques in defect detection of fruits and vegetables.

Product	Technique	Year	Type of Defect	Model	Accuracy	Reference
Apple	TRS	2014	Internal browning	LDA	90%, 71%	[94]
	SRS	2017	Mealiness	ANN	76%, 82%	[114]
	SIRI	2018	Fresh bruising	BDA	70%~100%	[83]
	SIRI	2018	Surface and Subsurface defects	CNN	98%	[85]
	SFDI	2021	Early bruising	ANN	MAE = 0.18%, 0.027% Rmse = 0.01, 0.14	[115]
	SFDI	2022	Early bruising	——	——	[112]
	SFDI	2022	Subsurface Bruising	cGANs	PSNR = 34.72 dB SSIM = 0.84	[116]
	SFDI	2023	Bruises	——	——	[113]
	SFDI	2023	Early bruises	CNN	two-category: 99.16% three-category: 91.50%	[117]
Pear	SFDI	2018	Bruising	Discriminant analysis	90%, 87.5%	[74]
	SFDI	2021	Surface damage	LDA	class 1: 92.5% class 2: 83.8%	[19]
	SFDI	2023	Bruising	LSTMR	MAE = 0.32%, 0.21%	[20]
Peach	SIRI	2019	Early decay	CNN	98.6%, 97.6%	[71]
	SRS	2020	Fungal infection	PLSDA	88%	[34]
	S-MSI	2023	Bruise detection	WSA	S1: 92% S2: 97.43% S3: 99.86%	[92]
Orange	SIRI	2022	Early decay	PLSDA	96.4%	[86]
Kiwifruit	SIRI	2023	Chilling injury	SVM	94.2%	[118]
Cucumber	SIRI	2021	Subsurface bruising	SVM	94%	[87]

ANN = artificial neural network, LDA = linear discriminant analysis, BDA = bruise detection algorithms, CNN = convolutional neural network, cGANs = conditional generative adversarial networks, PSNR = peak signal-to-noise ratio, SSIM = structural similarity index, MAE = mean absolute error, LSTMR = long short-term memory regressor, S-MSI = structured multispectral imaging, WSA = watershed algorithm.

**Table 6 foods-13-02688-t006:** Studies on the light penetration depth of light in fruits and vegetables.

Product	Technique	Year	Research Content	Result	Reference
Apple	SRS-MC	2009	Determine the optimum range of perception	*δ* is 0.43~8.67 cm over the 500~1000 nm; *δ* is related to wavelength	[31]
	MC	2015	Effects of skin thickness on light transport features in tissues	Thin skin has little effect on *δ*	[30]
	TRS	2020	Nondestructive assessment of the internal quality of fruits	*δ* is related to SDS	[60]
	SFDI-MC	2020 2021	Estimate $\mu_{a}$ and $\mu {’}_{s}$ of two-layered horticultural products	Smaller frequency could generate larger *δ*	[24,121]
Peach	MC	2015	Effects of the core and the skin on light transport features	Pericarp has more influence on *δ*	[122]
Citrus	——	2015	Effects of skin thickness on light transport features in tissues	Thick pericarp has great influence on *δ*	[30]
Melon	NIR	2018	Study the *δ* based on three SDS (10, 20 and 30 mm)	Depth ranges were 11.7~12.4, 14.2~15.8 and 16.8~21.0 mm	[123]
	VIS/NIR	2024	Investigate the light propagation through melons	*δ* is more than 2.5 mm in exocarp, 3 mm in green mesocarp and 3.5 mm in pulp	[55]
Pomelo	VIS/NIR	2022	Prediction of soluble solid content	*δ* determines the design of the detection system for large fruit	[33]
Persimmon	HSI-SR	2023	Estimate the *δ* of a HSI system in a Vis-NIR configuration (in 450~1050 nm)	*δ* is limited to about 2 mm	[32]
Turbid media	MC	2012	Influence of the illumination-collection geometry and optical properties on *δ*	*δ* is related to illumination-collection area and collection angle	[124]
	SRS-MC	2016	Estimate the optical properties of two-layer media	*δ* is proportional to SDS	[25]

*δ* = penetration depth. SDS = source-detector separation. NIR = near-infrared spectrum. SRS-MC = SRS coupled with MC. SFDI-MC = SFDI with MC.

## Data Availability

No new data were created or analyzed in this study. Data sharing is not applicable to this article.

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
