# Peer review of "Recent Advances in Light Penetration Depth for Postharvest Quality Evaluation of Fruits and Vegetables"

_foods, 2024, doi:10.3390/foods13172688_

Round 1

Reviewer 1 Report

Comments and Suggestions for Authors

The authors of the manuscript discusses recent advances in light penetration depth for postharvest quality evaluation of fruits and vegetables using emerging optical detection technologies. The review is well structured and clear, the paragraphs are related to each other, there is a fluent transition between the paragraphs.

However, I have some suggestions to make:

  • What is the aim of the review on light penetration depth?

  • What are the limitations of dielectric technique for quality assessment?

  • How is the light penetration depth determined in biological tissues?

Author Response

Q1What is the aim of the review on light penetration depth?

Answer: Thank you for your valuable comment. First of all, as one of the parameters for optical property, the light penetration depth has been studied by many researchers, but until now, no article has summarized these studies. Secondly, in line 75, most of the current studies consider fruits and vegetables as single homogeneous samples to simplify the optical transmission model, which will cause huge measurement errors for some fruits and vegetables with thicker pericarp, such as citrus and mangoes, have the great difference of pericarp and pulp tissues involving chemical composition and physical properties. While light penetration depth is helpful for accurate optical information in each tissue layer, which is conducive to improve the detection accuracy of fruit and vegetable quality evaluation. In addition, the light penetration depth could directly reflect the attenuation of light in tissues, which is significant to determine the best sensing position of the sample tissue so as to design more suitable devices for non-destructive detection of fruits and vegetables. Thus, the aim of this paper is to summarize the studies on light penetration to help the researches on accurate detection for multilayer- fruit and vegetables quality assessment along with better understanding the theory of light transmission.

Q2What are the limitations of dielectric technique for quality assessment?

Answer: Thank you for your valuable comment. The non-destructive detection technology based on dielectric characteristics could judge the firmness, maturity, sugar degree and moisture based on the dielectric constant, impedance, resistance and other electrical parameters of fruits and vegetables, which has broad prospects in sorting. However, the detection results are affected by the measurement frequency/voltage signal, moisture content, environmental temperature, humidity and other parameters. In addition, the commonly used parallel plate (capacitor) test system is often affected by interference factors such as boundary effect and individual difference of fruit shape, resulting in data drift and affecting the measurement accuracy, which limits the application of the dielectric technique.

Q3How is the light penetration depth determined in biological tissues?

Answer: Thank you for your valuable comment. In this paper, the determination of light penetration depth could be based on diffusion approximation equation and Monte Carlo method.

In the diffusion approximation equation, the light penetration depth could be determined according to Figure 3 and Equation (5). In line 87 and 190, according to Wilson's definition of penetration depth, when I/I0 = 1/e ≈37 %,δ = L . In line 94, according to Fraser's description, the light penetration depth is defined as distance reached when the light intensity drops to 1%, i.e., I/I0 = 1 %, here L = 4.6 δ,. In addition, under special conditions, it can be determined according to Equation 6 and 7.

In Monte Carlo method, the light penetration depth is the maximum longitudinal position reached by diffuse photons in tissues, as shown in Figure 4b. The results of light penetration depth are obtained by Monte Carlo simulation.

Reviewer 2 Report

Comments and Suggestions for Authors

Correspondence to the Author

The article under the title "Recent advances in light penetration depth for postharvest quality evaluation of fruits and vegetables" with the Manuscript ID foods-3150371 gives an improvement in the emerging technology for the prediction of food quality using a non-destructive methods and programs. The study holds the promise of being a significant reference within the field of pre- and postharvest study.

However, authors should respond to the comments below before publication.

1-L 53, “When illumination passes through biological tissues…”, authors should give an example for biological tissues. Should be stated what kind of biological tissues are examined in this review. This issue (biological tissue) is left hanging and general.

2- L25 and L 79, what do you mean “biological tissues or biological materials”. Is it used for fruits and vegetables or other living environments? “Biological materials, tissues” should be clarified by authors in the text.

3- L 79, If authors are using “biological tissue” for the living organisms, they should present a study from the literature.

3- L95, “SDS” Authors should always write the long name of a word in the text before using its abbreviation.

4- L 134, the authors should give a reference to Beer-Lambert law.

“I” could be explained as the transmitted light in equation. Molar extinction factor are given in this symbol “Ɛ”.

5- L144, Since Raman scattering is explained in these sections, authors should explain Stokes and Anti-Stokes scattering and should be shown and Raman and Rayleigh scattering also be explained with a figure.

6- In figure 1, what is the title of X-axis? It cannot be read. 

7- L 517, for application chapter, Although the authors provide some information about the product, the techniques studied in this field, the validation of these methods for assessing the quality of fruits or vegetables has not been discussed sufficiently throughout the entire text. The results, including the error rate and model accuracy for these methods, could be presented in a figure alongside each study. Additionally, it should be determined which technique is the best and closest to achieving high predictive accuracy. Furthermore, the most accurate results obtained for specific fruits and vegetables should be identified, and the reasons for the high accuracy of the model for these particular fruits or vegetables should also be explained.

8- L 556, In table 3, model WSA was obtained with the high accuracy rate for peach. This result specific to this fruit needs to be discussed.

9- In conclusion, the authors should provide the results indicating which optical detection methods yielded better outcomes and which ones had lower accuracy, validated with several samples. If all techniques demonstrate a similar success rate, this should also be mentioned in the conclusion.

10- There are some lacks in the manuscript text:

--Every definition should be written with long name and then its abbreviation could be used.

--The figure caps, the title axes should be checked.

--There are many errors such as punctuation and spacing in the entire text.

-- It would be beneficial to check the grammar.

Comments on the Quality of English Language

It would be beneficial to check the grammar.

Author Response

Q1L 53, “When illumination passes through biological tissues…”, authors should give an example for biological tissues. Should be stated what kind of biological tissues are examined in this review. This issue (biological tissue) is left hanging and general.

Answer: Thank you for your good suggestion. Biological tissues refer to the tissues of fruits and vegetables in the article. We have added relevant content in line 59.

Q2L25 and L 79, what do you mean “biological tissues or biological materials”. Is it used for fruits and vegetables or other living environments? “Biological materials, tissues” should be clarified by authors in the text.

Answer: Thank you for your valuable comment. In this paper, biological materials refer to fruit and vegetables. Based on your comments, we revised the content in line 25 and 85, and clarified “Biological tissues” in line 59.

Q3L 79, If authors are using “biological tissue” for the living organisms, they should present a study from the literature.

Answer: Thank you for your valuable comment. We have revised the content in line 85, and “biological tissue” refers to the tissues of fruits and vegetables in the review.

Q3L95, “SDS” Authors should always write the long name of a word in the text before using its abbreviation.

Answer: Thank you for your valuable comment. In line 84, we have added the abbreviation after the long name based on you comment.

Q4L 134, the authors should give a reference to Beer-Lambert law.

“I” could be explained as the transmitted light in equation. Molar extinction factor are given in this symbol “Ɛ”.

Answer: Thank you for your valuable comment. We have revised the content in line 139 and 141, and added corresponding reference. In the book named "Light Scattering Technology for Food Property, Quality and Safety Assessment", Equation 1 is simplified by Beer-Lambert law to be more suitable for the light transmission characteristics of fruit and vegetable samples.

Q5L144, Since Raman scattering is explained in these sections, authors should explain Stokes and Anti-Stokes scattering and should be shown and Raman and Rayleigh scattering also be explained with a figure.

Answer: Thank you for your valuable comment. Based on your comment, we have added the content and figure about Stokes and Anti-Stokes scattering from line 168 to 174, and explained the energy variation of Rayleigh scattering and Raman scattering (Stokes and Anti-Stokes) in Figure 2.

Q6In figure 1, what is the title of X-axis? It cannot be read.

Answer: Thank you for your valuable comment. We are so sorry about this mistake. The figure has been replaced in line 194.

Q7L 517, for application chapter, Although the authors provide some information about the product, the techniques studied in this field, the validation of these methods for assessing the quality of fruits or vegetables has not been discussed sufficiently throughout the entire text. The results, including the error rate and model accuracy for these methods, could be presented in a figure alongside each study. Additionally, it should be determined which technique is the best and closest to achieving high predictive accuracy. Furthermore, the most accurate results obtained for specific fruits and vegetables should be identified, and the reasons for the high accuracy of the model for these particular fruits or vegetables should also be explained.

Answer: Thank you for your valuable comment. According to the reviewer's comments, we have added the related content of discussion in the application chapter in line 579, 645, 692, 728, 814 and so on. Additionally, we also added the results of the corresponding studies, the accuracy of the model, and the evaluation indicators in Table 4 and Table 5. At the end of each section in application chapter, we added related content to discuss that each optical techniques are suitable for corresponding detection items. In addition, we also gave accurate results for specific fruits in line 610, 671, 677 and the reasons for the high accuracy of the mathematical models in line 648 and 732.

Q8L 556, In table 3, model WSA was obtained with the high accuracy rate for peach. This result specific to this fruit needs to be discussed.

Answer: Thank you for your valuable comment. We have added the corresponding discussion in line 711 to line 718 based on your suggestion.

Q9In conclusion, the authors should provide the results indicating which optical detection methods yielded better outcomes and which ones had lower accuracy, validated with several samples. If all techniques demonstrate a similar success rate, this should also be mentioned in the conclusion.

Answer: Thank you for your valuable comment. We have revised manuscript in line 579 to 581,line 650 to 653, line 728 to 732 and line 897 to 903 based on your suggestion.

Q10There are some lacks in the manuscript text:

--Every definition should be written with long name and then its abbreviation could be used.

--The figure caps, the title axes should be checked.

--There are many errors such as punctuation and spacing in the entire text.

-- It would be beneficial to check the grammar.

Answer: Thank you for your valuable comment. We have revised the content in line 84, 131 and 636 for the use of abbreviations. According to the reviewer's comments, we have corrected for figure caps, the title axes and errors such as punctuation and spacing in this manuscript.

Reviewer 3 Report

Comments and Suggestions for Authors

This review paper outlines the theory and principle of light penetration depth based on several emerging optical detection technologies, and to focus primarily on its applications in the field of quality evaluation of fruits and vegetables as well as its future applicability in biological materials along with the challenge in the future.

 However, according to my opinions, there are some following points that authors should consider again for the quality of their manuscript:

1)      The acronym SDS (Line 95) must be explicit at the first time. Please also check again for all acronyms used in the manuscript. They must be clarified at the first use.

2)       I would like to see a scheme of principles for optical approach in the introduction

3)       I would suggest having the matrix of applying methods to address the chronological trends reviewed, also if possible, the metrics for methods used?

4)      As this is a review, I would like to see a summary of comparison for all methods they reviewed, in which the positive points and negative points of each approach will be described.  

5)      As this is a review, cited references should open to more international authors and Journals, and more relevant to this work.

6)      The conclusion should be more elaborated.

Comments on the Quality of English Language

Acceptable. Please edit again before submitting.

Author Response

Q1The acronym SDS (Line 95) must be explicit at the first time. Please also check again for all acronyms used in the manuscript. They must be clarified at the first use.

Answer: Thank you for your good suggestion. We have added the abbreviation after the long name in line 84 and checked again for all acronyms, such as line 636.

Q2I would like to see a scheme of principles for optical approach in the introduction.

Answer: Thank you for your valuable comment. We have added schematic of the interaction between the light and object in Figure 1 since the optical detection technique is mainly based on the principle of interaction between light and biological tissues with the mode of light reflection, transmission and semi-transmission to obtain physical and chemical information. Moreover, the corresponding content was presented in line 56 to 59.

Q3I would suggest having the matrix of applying methods to address the chronological trends reviewed, also if possible, the metrics for methods used?

Answer: Thank you for your valuable comment. We have listed the related content in the Table 3~6  for fruit and vegetable in chronological order, and given the metrics in Table 4 and 5 along with the corresponding content in line 647.

Q4As this is a review, I would like to see a summary of comparison for all methods they reviewed, in which the positive points and negative points of each approach will be described. 

Answer: Thank you for your valuable comment. We have added Table 1 and 2 to compare the techniques and methods, the related content also presented in line 218 to 221 and 322.

Q5As this is a review, cited references should open to more international authors and Journals, and more relevant to this work.

Answer: Thank you for your valuable comment. We have added several relevant references in line 48,49 and 669 based on your suggestion. If you could provide other references, we are looking forward to your recommendation.

Q6The conclusion should be more elaborated.

Answer: Thank you for your valuable comment. We have added more detailed content in the conclusion section, including the definition and calculation of the light penetration depth along with the discussion of the corresponding optical detection techniques.
